# Impact of Runner Size, Gate Size, Polymer Viscosity, and Molding Process on Filling Imbalance in Geometrically Balanced Multi-Cavity Injection Molding

**DOI:** 10.3390/polym16202874

**Published:** 2024-10-11

**Authors:** Minyuan Chien, Yaotsung Lin, Chaotsai Huang, Shyhshin Hwang

**Affiliations:** 1Department of Vehicle Engineering, Chien Hsin University of Science and Technology, Taoyuan City 320678, Taiwan; jackyaren@uch.edu.tw; 2Graduate Institute of Precision Manufacturing, National Chin-Yi University of Technology, Taichung 411030, Taiwan; train@ncut.edu.tw; 3Department of Chemical and Materials Engineering, Tamkang University, New Taipei City 251301, Taiwan; cthuang@mail.tku.edu.tw; 4Department of Mechanical Engineering, Chien Hsin University of Science and Technology, Taoyuan City 320678, Taiwan

**Keywords:** filling imbalance, multi-cavity, microcellular injection molding, polymer, composites, shear rate

## Abstract

The injection molding process is one of the most widely used methods for polymer processing in mass production. Three critical factors in this process include the type of polymer, injection molding machines, and processing molds. Polypropylene (PP) is a widely used semi-crystalline polymer due to its favorable flow characteristics, including a high melt flow index and the absence of a need for a mold temperature controller. Additionally, PP exhibits good elongation and toughness, making it suitable for applications such as box hinges. However, its tensile strength is a limitation; thus, glass fiber is added to enhance this property. It is important to note that the incorporation of glass fiber increases the viscosity of PP. Multi-cavity molds are commonly employed to achieve cost-effective and efficient mass production. The filling challenges associated with geometrically balanced layouts are well documented in the literature. These issues arise due to the varying shear rates of the melt in the runner. High shear rate melts lead to high melt temperatures, which decrease melt viscosity and facilitate easier flow. Consequently, this results in an imbalanced filling phenomenon. This study examines the impact of runner size, gate size, polymer viscosity, and molding process on the filling imbalanced problem in multi-cavity injection molds. Tensile bar injection molding was performed using conventional injection molding (CIM) and microcellular injection molding (MIM) techniques. The tensile properties of the imbalanced multi-cavity molds were analyzed. Flow length within the cavity served as an indicator of the filling imbalance. Additionally, computer simulations were conducted to assess the shear rate’s effect on the runner’s melt temperature. The results indicated that small runner and gate sizes exacerbate the filling imbalance. Conversely, glass fiber-filled polymer composites also contribute to increased filling imbalance. However, foamed polymers can mitigate the filling imbalance phenomenon.

## 1. Introduction

Most modern plastic parts are currently manufactured using an injection molding process. A common method for cost reduction in manufacturing is to increase production capacity, which can be achieved through multi-cavity molds. Initially, multi-cavity molds were designed in a fishbone or family-type configurations. While these designs can reduce mold size and cost, they are prone to filling imbalance issues during injection. This imbalance is caused by high shear rate regions near the mold wall [1,2] on the sprue. The high-shear region, which is hotter than the low-shear region, leads to a nonlinear temperature distribution along the runner [3]. When the polymer melt makes a 90-degree turn from the sprue to the first runner, the hotter melt is present on one side, while the colder melt is on the opposite side, as illustrated in Figure 1. Temperature variations influence the viscosity of the polymer melt. As a result, a filling imbalance occurs when the polymer enters the gate, leading to quality issues in the final part [4]. Through the advance of computer technology, computer simulation of injection molding is widely used. Researchers or manufacturers can save time in molding manufacture or defect prediction by computer simulation. García-Granada et al. [5] used Polyflow software to solve nanopattern replication of complex structures by injection molding.

Later, cavities were modified to achieve geometrical balance [1], where the distance from the sprue to the gate is equal across all cavities. Despite this improvement, geometrically balanced layouts still encounter filling imbalance issues. Yang et al. [3] investigated an eight-cavity geometrical balance mold with H-type runners, examining the effects of injection speed, melt temperature, and mold temperature on filling imbalance. Their findings indicated that adjusting these three process parameters could reduce filling imbalance. Subsequently, Tsai et al. [6] employed a four-cavity mold for the optical lens and introduced a flow limiter in the runner to induce turbulent flow. This approach aimed to achieve a more uniform melt temperature across the cavities, thereby reducing residual stress and shrinkage/warpage. The key factor in the flow limiter was its depth, which influenced temperature distribution. Additionally, the adoption of the flow limiter resulted in decreased lens roughness. Beaumont et al. [7] addressed the filling imbalance problem using the melt flipper^®^ technique for multi-cavity molds. They found that cavities with high shear rates exhibited increased crystallinity and Young’s modulus. Wilczyńsk et al. [8] explored the filling imbalance issue in geometrically balanced injection molding through both experimental and simulation approaches in 2018. They tested various runner designs and process parameters to mitigate filling imbalance but noted that it could not be completely eliminated. The discrepancy between the experimental and simulation results increased with greater imbalance. Their continued research included mold flow simulations to evaluate the effects of material properties and runner design, and they concluded that factors such as heat transfer coefficient, thermal diffusivity, shear stress, index, and polymer viscosity strongly influenced filling imbalance [9]. They also employed several optimization techniques, including artificial neural networks, the Taguchi method, and response surface methodology, to improve the imbalance problem by experimentation and address the imbalance issue through experimentation and mold flow simulation [10,11]. The study targeted different runner systems under varying polymer viscosities and process conditions, with artificial neural networks proving to be the most effective. The filling imbalance problem also extends to metal power injection molding (MPIM) [12], which requires a longer filling time compared to polymer injection molding due to lower injection speeds. Rhee et al. [13] utilized pressure and temperature sensors to monitor pressure and temperature profiles on the runners of a multi-cavity mold for a small lens. Their results indicated that the temperature profile correlated with the melt-arrival time, whereas the pressure profile did not, confirming that temperature variations contribute to filling imbalance.

Assuming polymer melt is a general Newtonian fluid, the non-isothermal flow can be described by the following equation [14]:(1)$$\frac{\partial \rho }{\partial t}+\nabla \cdot \partial u=0$$
(2)$$\frac{\partial }{\partial t}\left(\partial u\right)+\nabla \cdot \left(\partial uu-\sigma \right)=\rho g$$
(3)σ=−PI+τ
(4)$$\rho Cp\left(\frac{\partial T}{\partial t}+u\cdot \nabla T\right)=\nabla \cdot \left(k\;\nabla T\right)+\tau :D$$
where *ρ* is the density, *u* is the velocity vector, *σ* is the total stress tensor, *g* is the gravity, *P* is the pressure, *η* is the viscosity, I is the identity, *T* is the temperature, *t* is the time, *k* is the thermal conductivity, *Cp* is the specific heat, *τ* is the extra tensor, and *D* is the rate of deformation tensor.
(5)τ=2 η D

The temperature-dependent viscosity of a polymer melt based on the modified Cross model is given by:(6)$$\;\;\;\;\eta \left(T,\;\dot{\gamma }\right)=\frac{\eta_{0}(T)}{1+{\left(\eta_{0}\dot{\frac{\gamma }{\tau^{*}}}\right)}^{1-n}}$$
(7)$$\eta_{0}\left(T\right)=B\;Exp\frac{T_{b}}{T}$$
where $\dot{\gamma }$ is the shear rate, $\eta_{0}$ is the zero shear viscosity, *n* is the power law index, and $\tau^{*}$ is the parameter that describes the transition between the zero-shear region and the power-law region of the viscosity curve.

Conventional injection molded (CIM) parts often exhibit defects such as shrinkage/warpage due to non-uniform thickness [15]. To address these issues, microcellular injection molding (MIM) was introduced to overcome the shrinkage/warpage problem [16,17] and provide benefits such as material savings, weight reduction, and reduced cycle time benefits [18]. Consequently, MIM is considered a more environmentally friendly manufacturing process. During CIM, the polymer melt is compressed within the mold, whereas in MIM, the polymer melt is expanded. This difference leads to distinct filling patterns (i.e., temperature distribution) within the runner.

To date, few studies have examined the effects of polymer types, runner and gate sizes, and molding methods (such as CIM and MIM) on the filling imbalance problem in injection molding. This study employed both CIM and MIM processes to investigate the filling imbalance phenomenon. The test specimen was an ASTM standard D 638 [19] type tensile bar. Two geometrical factors—runner size and gate size (fan gate)—were evaluated for their potential impact on filling imbalance. Specifically, runner sizes of 3 and 6 mm and gate sizes of 2.03 and 7.42 mm^2^ were tested. Both conventional and microcellular injection molding techniques were used to produce the tensile bar. Due to filling imbalance issues, some cavities were over-packed compared to others. A tensile test was conducted to assess the tensile properties of the samples. Unfilled or filled polymers with varying viscosities were examined to evaluate the viscosity effect. To date, there has been limited research on the impact of runner/gate size, polymer viscosity, and molding method on the filling imbalance in geometrically balanced molds.

## 2. Materials and Methods

### 2.1. Material

This study utilized polymers with varying viscosities, including polypropylene (PP), polycarbonate (PC), glass fiber-filled PP, and glass fiber-filled PC. PC exhibits higher viscosity than PP, resulting in easier flow for PP melt than PC melts. PP (ST868M) was supplied by LCY Group Chemical Co., Ltd., Taipei, Taiwan, while Chi-mei Co., Ltd., Tainan, Taiwan, supplied PC (PC122U).

### 2.2. Mold

The mold comprises eight cavities arranged in a geometrically balanced layout. It features two different runner sizes (3 and 6 mm) and two fan gate sizes (2.03 and 7.42 mm^2^), as shown in Figure 2. The length from the sprue center to the end of the first runner measures 35 mm, with a diameter of 7 mm at the end of the sprue. The length of the third runner is 18 mm. The cavity shape is a tensile bar, as specified by the ASTM D 638 standard illustrated in Figure 3. Ten test samples were collected to measure the short shot flow length, with ten moldings obtained for each polymer molding. The flow length measurement was taken from the gate to the maximum distance, although the shape of the melt front was not regulated. The flow length deviation was calculated by subtracting the average flow length from the measured value using Excel 2019 software.

### 2.3. Injection Molding Machine

The polymers—PP, PC, 30 wt% glass fiber-filled PP, and 30 wt% glass fiber-filled PC composites—were molded using both conventional and Mucell^®^ injection molding processes (Trexel, Boston, MA, USA). The Mucell^®^ injection process was carried out on a 100-ton ARBURG-420C injecting machine equipped with a microcellular injection molding capability. Nitrogen (N_2_) was used as the physical blowing agent. The process parameters are detailed in Table 1. The melt temperature for the 30 wt% glass fiber-filled polymers typically requires a 20 °C increase compared to that of the neat polymers. For the tensile test, both CIM and MIM samples were allowed to stabilize for one week, enabling the solid and foamed samples to achieve stability before testing.

### 2.4. Instrumentation

The tensile test of the ASTM standard sample was conducted using a tensile test analyzer, HT-9102M, manufactured by Hong-Da Company, Taichung, Taiwan. For PP, the axial speed was set at 1000 mm/s. Different polymers and composite samples required different axial speeds, with composites typically having lower axial speeds compared to neat polymers. According to ASTM D638 standards, the tensile test should last at least 120 s. Capillary viscosity measurements were performed using a Smart RHEO rheometer from CEAST, Italy, with a capillary diameter of 1 mm. The rheometer provided apparent viscosity measurements over a shear rate range from 100~10,000 1/s. Melt flow simulations were conducted using Moldex3D R21 moldflow software provided by CoreTech, Hsinchu, Taiwan.

## 3. Results and Discussion

The following results are organized into several subsections: short shot flow for different runner sizes and gate sizes for neat and glass fiber-filled polymers, followed by a discussion of tensile strength and polymer viscosity. Finally, a comparison between experimental results and computer simulations is presented. The runner and gate sizes must be appropriately designed; otherwise, poor design can lead to issues after cooling the sample from a design point of view [15].

### 3.1. Short Shot Flow Length

It is preferable to perform short shot molding without packing during the initial molding of trials of a new mold or polymer. The shot size should be gradually increased until the cavity is completely filled. Visual inspections of the molded parts are conducted to assess the effects of increasing shot size. Appropriate measures were then taken to address any issues in the filling process. Initially, a 3 mm runner with a small gate was custom-made to investigate flow length comparisons. However, using small and big gates on the 6 mm runner was also considered, eliminating the need to change the mold insert. Thus, small gates were positioned on the upper side and big gates on the lower side.

#### 3.1.1. Short Shot Flow Length for 3 mm Runner

Figure 4 illustrates the short shot flow length of a 3 mm runner for PP and glass fiber-filled PP using both conventional and microcellular injection molding techniques. Conventional molded PP and glass fiber-filled PP are shown on the left-hand side (Figure 4a). These samples exhibit longer flow lengths in cavities #2 and #3. Despite the balanced layout, a filling imbalance problem persists, primarily due to the high shear rate at the sprue. The flow length for glass fiber-filled PP is longer than that of neat PP, attributed to higher temperatures in the high shear rate region caused by the interaction between PP and glass fibers [20,21].

Figure 5 presents the short shot flow length for a 3 mm runner for conventional-molded PC and microcellular injection-molded PC. With its higher viscosity compared to PP [22], PC results in shorter flow lengths. The variation in flow length is more pronounced for PC due to its higher viscosity. On the right-hand side, a MIM molded PC exhibits longer flow lengths due to cell expansion during foaming [18]. A detailed examination reveals that the melt front profiles in cavities #2 and #3 differ from those in cavities #1 and #4 for CIM PC (left-hand side). Notably, the left-hand side shows longer flow lengths than the right-hand side of cavity #3. This observation is consistent with Wilczynski’s study [9]. The hot melt tends to be biased to the left-hand side, a phenomenon not observed in the MIM parts (right-hand side). Foaming alters the melt flow and temperature distribution [23].

#### 3.1.2. Short Shot Flow Length for a 6 mm Runner

Figure 6 illustrates the short shot flow length for a 6 mm runner in conventional injection molded PP and glass fiber-filled PP. The upper cavities feature small gates, while the lower cavities have big gates. The big gate (with a cross-section area of 7.42 mm^2^) exhibits a longer flow length compared to the small gate (2.03 mm^2^). Despite the common understanding that big runners or gates generally yield better parts quality, this concept is not widely accepted in the industry. Manufacturers often prefer small gates and runners to expedite production, as reduced cycle time enhances profitability [24,25]. 

**Figure 6 polymers-16-02874-f006:**
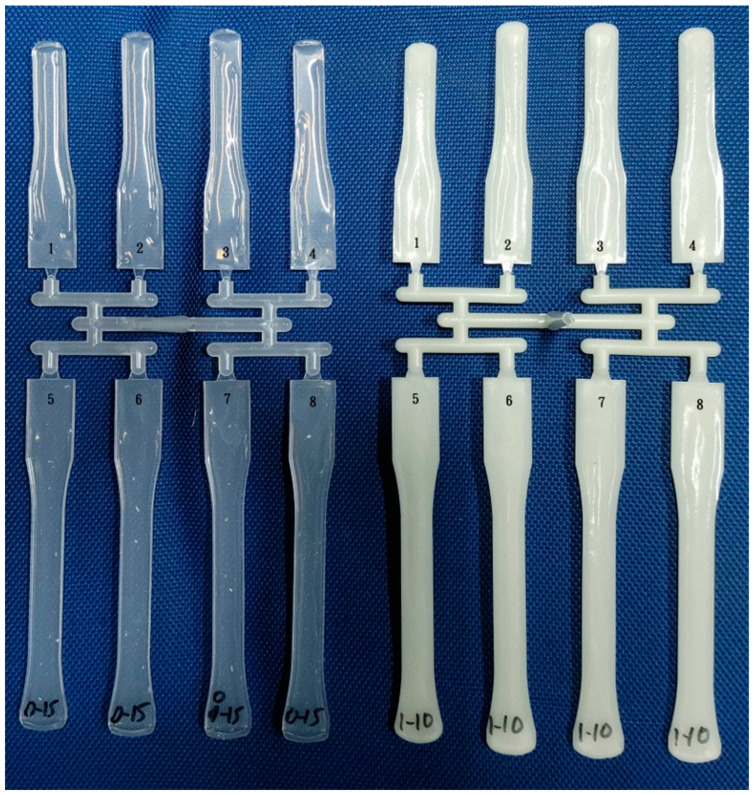
Short shot flow length of 6 mm runner of conventional injection molded PP and glass fiber-filled PP. Upper cavities are small gates, and lower cavities are big gates. Number 1–8 is the cavity number.

#### 3.1.3. Short Shot Flow Length Comparison for a 6 mm Runner

Short shot flow lengths for PP, PC, glass fiber-filled PP, and glass fiber-filled PP were measured on a 6 mm runner, with cavities either featuring small gates (cavities #1~#4) or big gates (cavities #5~#8). Figure 6, Figure 7, Figure 8, Figure 9, Figure 10, Figure 11, Figure 12 and Figure 13 present the flow length deviations, which are summarized in Table 2 for comparison.

Figure 7a compares the flow length between conventional injection molded PP and glass fiber-filled PP using a 6 mm runner and a small gate. The flow length deviations for neat PP and glass fiber-filled are 5.35 and 10.5 mm, respectively. The glass fiber-filled PP exhibits a shorter flow length than the neat PP with a big runner and a small gate. This contrasts with observations for small runners and small gates (Figure 3), where the undersized gate adversely affects the injection process [26]. Figure 7b shows the flow length comparisons between conventional injection molded PP and glass fiber-filled PP using a 6 mm runner and a big gate. The big gate facilitates more polymer melts into the cavity, resulting in flow length deviations of 1.3 and 3.8 mm for neat PP and glass fiber-filled PP, respectively. Moreover, the flow length of glass fiber-filled PP is longer than that of neat PP. In addition, both PP and glass fiber-filled PP have flow lengths (~14 cm) longer than those of samples with small gates (~9 cm). Smaller runners require higher injection pressure to force the polymer melt into the cavity, potentially trapping air in the molded part and leading to defects [26,27].
Figure 7Flow length comparisons between conventional injection molded PP and glass fiber-filled PP on 6 mm runner: (**a**) small gate and (**b**) big gate.
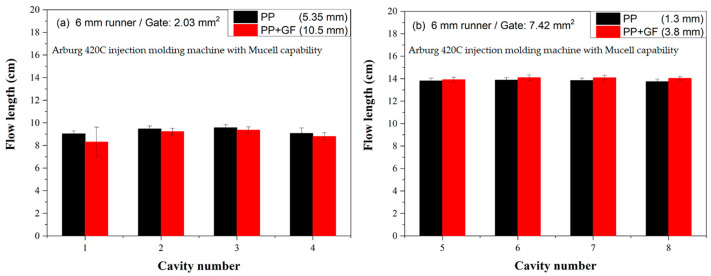


Figure 8a compares the flow length between microcellular injection-molded PP and glass fiber-filled PP using a 6 mm runner and a small gate. PP showed longer flow lengths for small gates than those of glass fiber-filled PP for MIM molded parts. The flow length deviations are 4.95 and 6.25 mm for neat and glass fiber-filled PP, respectively, which are smaller than the deviations observed in conventional injection molded samples of 5.35 and 10.5 mm, respectively.

Figure 8b compares the flow lengths of microcellular injection-molded PP and glass fiber-filled PP using a 6 mm runner and a big gate. The flow lengths followed a similar trend to that observed with small gates. The flow length of glass fiber-filled PP was shorter than that of neat PP due to the reduced influence of cell expansion growth compared to the viscosity effect.
Figure 8Flow length comparisons between microcellular injection molded PP and glass fiber-filled PP on a 6 mm runner: (**a**) small gate and (**b**) big gate.
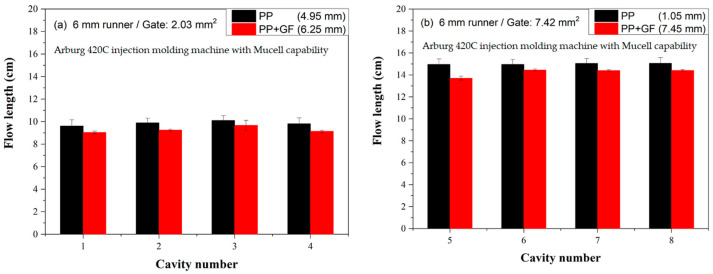


Figure 9a compares the flow lengths of conventional injection-molded PC and glass fiber-filled PC using a 6 mm runner and a small gate. PC is widely used in optical applications, such as eyeglasses, due to its toughness (compared to polystyrene and polymethylmethacrylate). However, it has poor flowability [28]. Consequently, the flow length deviations are substantial for both PC and glass fiber-filled PC, with deviations of 26.15 and 11.35 mm, respectively. However, the length deviation of a glass fiber-filled PC was less than that of a neat PC.

Figure 9b presents flow length comparisons between conventional injection-molded PC and glass fiber-filled PC using a 6 mm runner and a big gate. The flow length of the glass fiber-filled PC was longer than that of the neat PC on the big gate, primarily due to the easier flow in the big gate than that in the small gate. The length deviation of a glass fiber-filled PC was 0.7 mm, which is less than that of a neat PC (4.75 mm).
Figure 9Flow length comparisons between conventional injection molded PC and glass fiber-filled PC on 6 mm runner: (**a**) small gate (**b**) big gate.
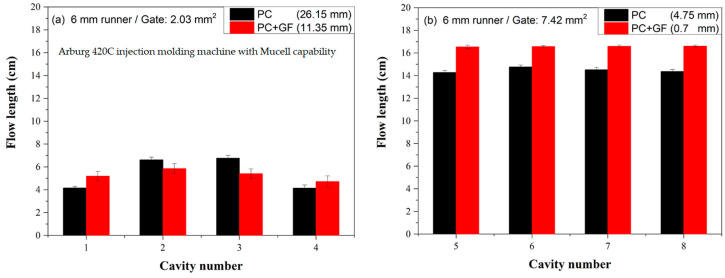


Figure 10a compares flow lengths of microcellular injection-molded PC and glass fiber-filled PC using a 6 mm runner and a small gate. Due to the high viscosity of the PC, the flow length for the neat PC was short (less than 2 cm), whereas the flow length for a glass-filled PC was twice than that of the neat PC. This might be caused by the high shear rate on the small gate causing glass fiber filled PP to be hotter than that of neat PC. The flow length deviation for glass fiber-filled was larger than that of neat PC.

Figure 10b compares the flow lengths of microcellular injection-molded PC and glass fiber-filled PC using a 6 mm runner and a big gate. The flow lengths of the neat PC and glass fiber-filled PC in the big gate were greater than those in the small gate. A big gate could supply sufficient polymer melt into the cavity. The flow length deviation for glass fiber-filled was less than that of neat PC. So, there is no regular rule for the length deviation of MIM molded PC and its composites, as shown by comparing Figure 13 and Figure 14.
Figure 10Flow length comparisons between microcellular injection molded PP and glass fiber-filled PP on a 6 mm runner: (**a**) small and (**b**) big gate.
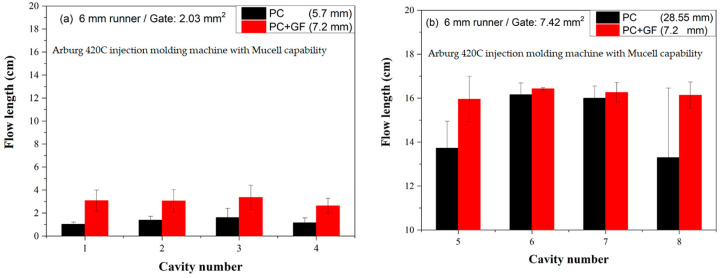


For easy comparison, the flow length deviations from Figure 7, Figure 8, Figure 9 and Figure 10 are summarized in Table 2. There appears to be a conclusive trend in conventional injection molding, where small gates are associated with large deviations and big gates with small deviations. However, this trend does not hold for MIM, where no regular relationship between small and big gates is observed. Typically, the weight of parts produced by the CIM can be maintained at a certain value (e.g., 1 wt%), while the weight of parts produced by the MIM process—which is more complex—varies significantly, sometimes reaching as much as 5 wt%, based on our observations. Several parameters can influence cell growth (e.g., injection speed and packing pressure). Although these parameters were kept constant, weight variations remain substantial in the MIM process, resulting in an irregular relationship between small and big gates.

### 3.2. Polymer Viscosity

Figure 11 shows the apparent viscosity of PP, PC, glass fiber-filled PP, and glass fiber-filled PC as measured by capillary rheometer. The measurements were taken at 210 °C for PP, 220 °C for PP + GF, and 320 °C for both PC and PC + GF. The glass fiber-filled PP exhibits the highest viscosity at a low shear rate, ranging from 100 to 400 1/s. The glass fiber-filled PP and fiber-filled PC have higher viscosities than their corresponding neat polymers [21]. However, neat and glass fiber-filled PC exhibit higher viscosities at a high shear rate than PP [28], with a significant difference observed.
Figure 11Apparent viscosity of PP, PC, glass fiber-filled PP, and glass fiber-filled PC by capillary rheometer. PP, PP + GF, PC, and PC + GF were measured at 210, 220, 320, and 320 °C respectively.
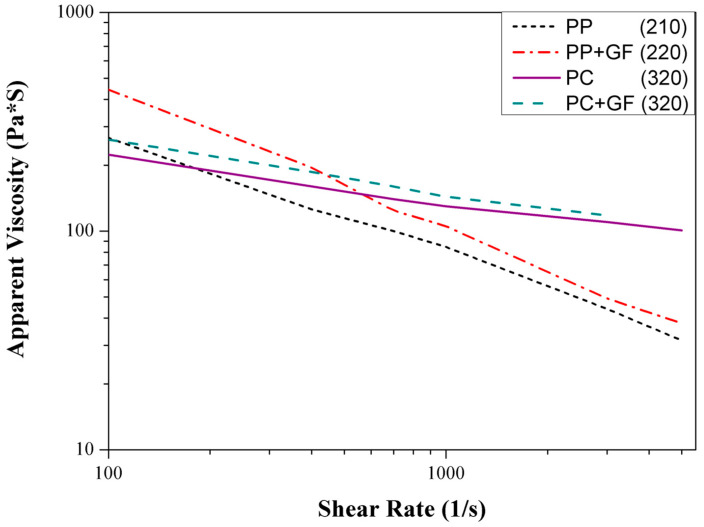


### 3.3. Tensile Strength

Out of curiosity, the effect of filling imbalance on the tensile strength of the molded sample was investigated. Consequently, a full shot of PP and PP + GF was employed. Figure 12 shows the tensile strength comparison between PP and glass fiber-filled PP on a 3 mm runner by conventional molding. Figure 12a illustrates the tensile strength of neat PP. The tensile strength increases to a maximum value before decreasing to a plateau, indicating that necking occurs, which is associated with excellent elongation at break. Samples from cavities #1 to #4 have almost the same tensile strength, but cavities #2 and #3 have better elongation (280%) compared to cavities #1 and #4 (125%). The polymer’s tensile strength depends on the polymer chain’s orientation. During filling, highly oriented polymer chains result in better tensile strength [29,30]. Moreover, the better elongation observed in cavities #2 and #3 was caused by the over-packing of the polymer chain during the filling stage. Glass fiber-filled PP’s tensile strength (Figure 12b) is almost twice that of the neat PP, though its elongation is significantly lower. This observation agrees with the result of the long holding time by gas counter pressure molding [31]. The tensile strength increment in the composites is a function of the glass fiber [32].
Figure 12Tensile strength of (**a**) neat PP and (**b**) glass fiber-filled PP on a 3 mm runner by conventional molding.
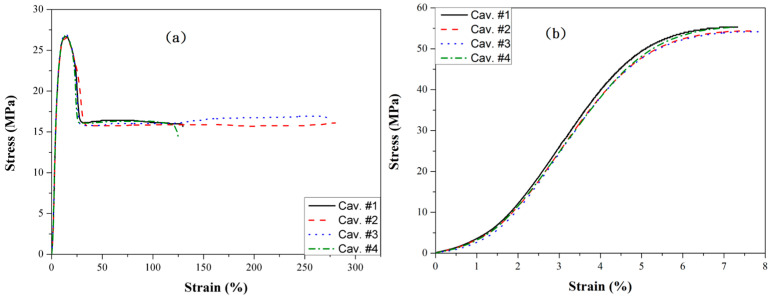


Figure 13 presents the tensile strength of (a) neat PP and (b) glass fiber-filled PP on a 6 mm runner with a small gate using the MIM process. Compared to Figure 12, which depicts results from the CIM process, the tensile strengths are reduced to approximately 27% of those achieved by CIM, while the elongation at break remains nearly unchanged. Cavities #1 to #4 exhibit similar elongation at the break due to the inertial pressure, which is around 10 MPa, being less than the packing pressure of 60 MPa. Consequently, cavities #2 and #3 do not experience overpacking.
Figure 13Tensile strength of (**a**) neat PP and (**b**) glass fiber-filled PP on a 6 mm runner and small gate by MIM.
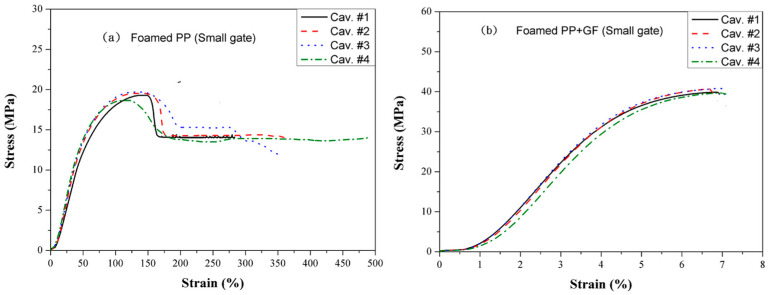


The tensile strengths from cavities #5 to #8 of the big gate and 6 mm runner using the MIM process are presented in Figure 14. Compared to Figure 13, which shows results from a small gate, the tensile strengths increase from 19 MPa to 22 MPa for cavities #1 to #4 of neat PP, while tensile strengths increase from 41 MPa to 49 MPa for glass fiber-filled PP on the 6 mm runner and big gate using MIM. This enhancement in tensile strength can be attributed to the big gate’s ability to supply sufficient polymer to fill the cavity completely.
Figure 14Tensile strength of (**a**) neat PP and (**b**) glass fiber-filled PP on a 6 mm runner and big gate by MIM.
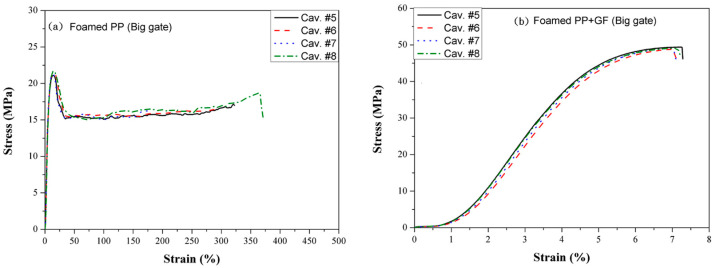


### 3.4. Moldflow Analysis

Moldflow analysis was performed to understand the temperature profile of the different runner sections using the shear rate of CIM molding. The temperature-dependent polymer viscosity used was the modified Cross model, as shown in Equation (6). Figure 15 shows the temperature profile of a 3 mm runner of glass fiber-filled PP. Figure 15 depicts the mold flow analysis of the temperature distribution of the 3 mm runner of glass fiber-filled PP. The hot temperature profile is close to the second runner’s sprue (mold center). This high shear rate on the sprue [2] is shown in Figure 1. Subsequently, when it makes a 90-degree turn on the third runner, the hot melt flows into cavities #2 and #3, making their temperatures higher than those of cavities #1 and #4. The heavy color represents the high temperature of the runner. Thus, cavities #2 and #3 had longer flow lengths than cavities # 1 and #4, which agrees with the experimental results for CIM PP and PC.
Figure 15Mold flow analysis of the temperature distribution of the 3 mm runner of glass fiber-filled PP. The maximum scale of dark red color is 236 °C.
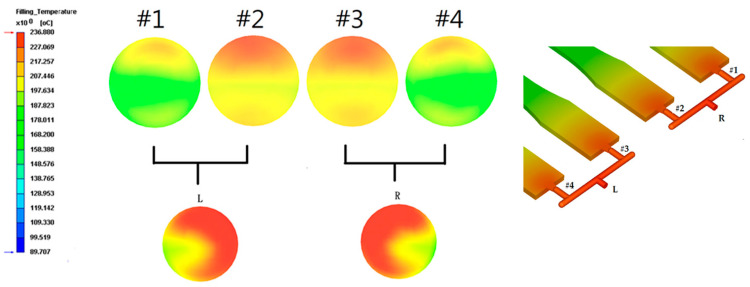


## 4. Conclusions

This study investigated the filling imbalance problem in a geometrically balanced layout. The variable parameters included runner size, gate size, polymer viscosity, and molding method. Several conclusions can be drawn from the results presented above and the discussion regarding flow length deviation, tensile strength variations among the four cavities, polymer viscosity, and the temperature profile of the runner.

The big gate exhibited a longer flow length than the small gate for both the CIM and MIM processes. For the flow length deviation, neat PP demonstrated a slight flow length deviation compared to the glass fiber-filled PP for both the CIM and MIM processes. However, this pattern does not hold for PC and glass fiber-filled PCs in either the CIM or MIM processes.Glass fiber-filled polymers exhibited higher viscosity than neat polymers for both PP and PC.Cavities experiencing over-packing situations demonstrated better elongation for both neat PP and glass fiber-filled PP in the CIM process, while the tensile strengths remained the same for types of PP.The results of moldflow analysis agree with the experimental results, indicating that moldflow analysis can reduce costs and mold manufacturing time, thereby reducing the preparation time for the injection molding process.Polymer foaming can disturb the temperature distribution of the polymer melt in the runner for the multi-cavity geometrically balanced mold.

## Figures and Tables

**Figure 1 polymers-16-02874-f001:**
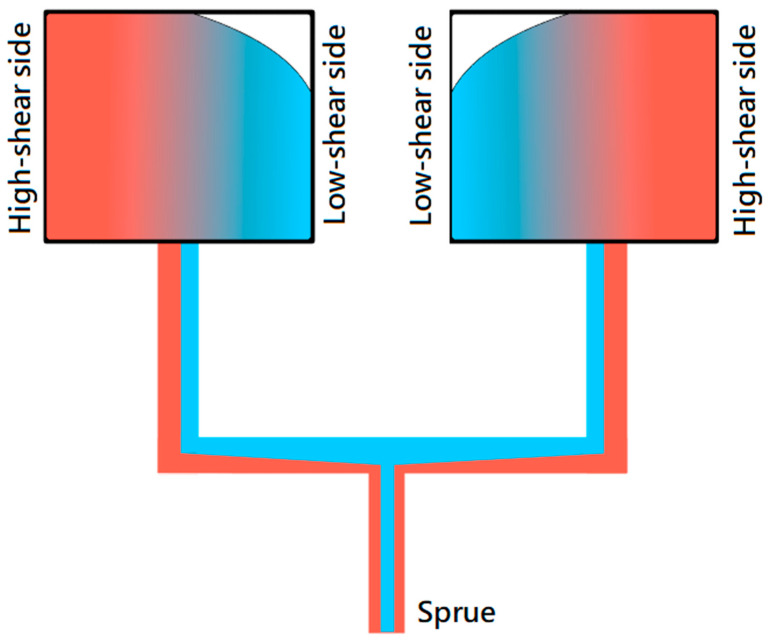
The effect of low shear rate and high shear rate on the melt temperature of the runner.

**Figure 2 polymers-16-02874-f002:**
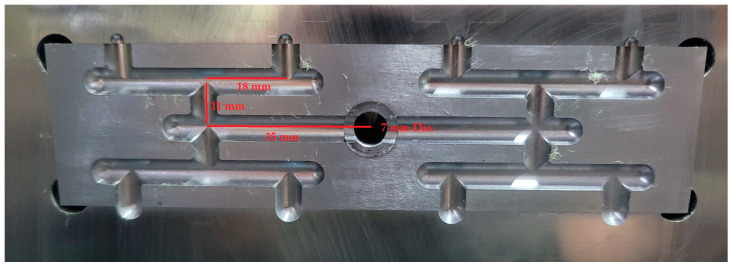
6 mm runner and small/big fan gate insert on the cavity.

**Figure 3 polymers-16-02874-f003:**
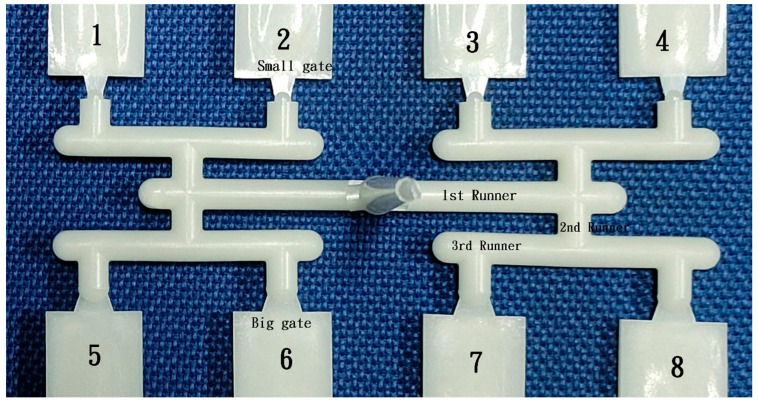
6 mm runner and gate system for geometrically balanced eight-cavity mold layout. Number 1–8 is the cavity number.

**Figure 4 polymers-16-02874-f004:**
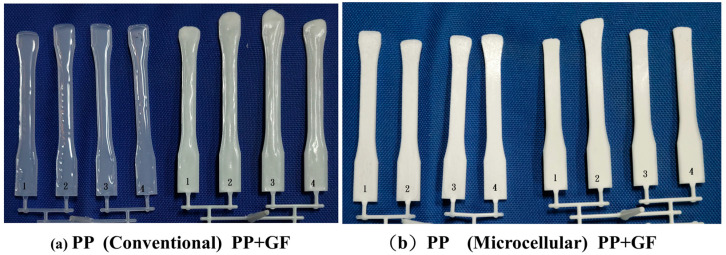
Short shot flow length of a 3 mm runner of PP, glass fiber-filled PP, conventional and microcellular injection molding. Number 1–4 is the cavity number.

**Figure 5 polymers-16-02874-f005:**
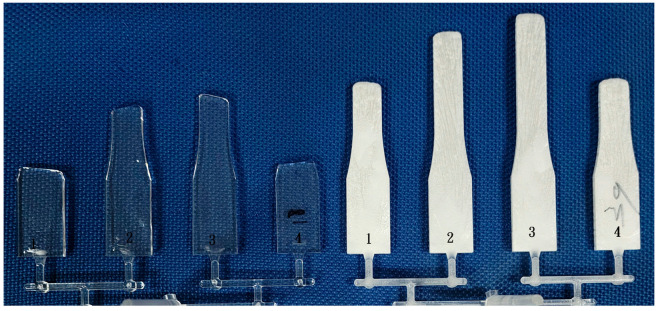
Short shot flow length of a 3 mm runner of conventional molded PC and microcellular injection molded PC. Number 1–4 is the cavity number.

**Table 1 polymers-16-02874-t001:** Process parameters for CIM and MIM.

Process Parameter	Conventional	Microcellular
Melt temp. for PP (°C)	210	210
Melt temp. for PC (°C)	310	310
Shot size (mm)	55	50
Screw speed (mm/s)	25	25
Injection speed (mm/s)	100	100
Injection P. (Bar)	1100	1100
Mold temp. for PP (°C)	50	50
Mold temp. for PC (°C)	90	90
Packing P. (Bar)	600	
Back P. (Bar)		100
N_2_ (%)		0.4

**Table 2 polymers-16-02874-t002:** Flow length deviation comparisons for CIM and MIM PP, PC, PP + GF, and PC + GF.

Polymer	Gate Size (cm^2^)	Length Deviation (mm)	Molding
PP	2.03	5.35	CIM
	7.42	1.3	
PP + GF	2.03	10.5	CIM
	7.42	3.8	
PP	2.03	4.98	MIM
	7.42	1.15	
PP + GF	2.03	6.25	MIM
	7.42	7.45	
PC	2.03	26.15	CIM
	7.42	4.75	
PC + GF	2.03	11.35	CIM
	7.42	0.7	
PC	2.03	5.7	MIM
	7.42	28.55	
PC + GF	2.03	7.2	MIM
	7.42	7.2	

## Data Availability

The data presented in this study are available on request from the correspondence author.

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
