# Peer review of "Impact of Runner Size, Gate Size, Polymer Viscosity, and Molding Process on Filling Imbalance in Geometrically Balanced Multi-Cavity Injection Molding"

_polymers, 2024, doi:10.3390/polym16202874_

Round 1

Reviewer 1 Report

Comments and Suggestions for Authors

Summary: As the title suggest the authors study Impact of Runner Size, Gate Size, Polymer Viscosity, and 2 Molding Process on Filling Imbalance in Geometrically balanced Multi-cavity Injection molding. The authors use PP and PC and their mixtures with glass fiber to study the differences in flow length and flow length deviation. The authors have designed a custom-made mold to produce ASTM D638 samples for analysis. Later on in this study they also use simulations and modelling to explain their results.

Overall Thoughts:  The authors have presented the results and discussion in a very clear manner. Most conclusions are supported by data very over-speculation. However, I have following suggestions for the authors to improve the current manuscript.

Suggestions/Correction:

Abstract. Remove line 16 and replaced imbalanced in line 18 with balanced as the context in this part of the abstract is extremely confusing.

Introduction. The introduction provides enough background to the current work done; however, it fails to identify the current knowledge gap and hence the need for the current study. I recommend separating the knowledge gap out in a separate paragraph and then have a paragraph outlining the work in this study.

Methods. Please add info on how the flow length were measure and how the Flow length deviations were calculated.

Results and Discussion.

-        Line 227, I think the authors are referring to figure 4.a here. Several other instances of incorrect mentions of the figure number were observed. Please rectify this.

-        Figure 7- 14. Please also label with the type of injection molding was used, below the runner length and gate size.

-        I would suggest pairing the figures with similar conditions but different gate size. For example, figure 7 as 7.a and Figure 8 as 7.b. This will allow better comparison between the gate size parameter and its effects. Please ensure the Y-axis of all the figures are same to allow better visual comparison.

-        Line 256 and 257, the PP needs to be edited properly.

-        Line 310, Add explanation on why the trend doesn’t hold true for MIM.

-        Line 331, 338 à it needs to be 16 a, b

-        Figure 17, add a gradient scale for what each color means

Conclusions. The authors need to re-write conclusions.  The current set of conclusions are very generalized and certainly do not point out the various non-linear and novel observations from this study. Additionally, the conclusions do not indicate what the future studies should explore to further the current work.

References.  Consistent formatting is recommended. Some references are missing page numbers and in some cases DOI is missing. It is up to the authors, what needs to be included however, the format should be consistent across all the references.

Author Response

(I) Ensure all references are relevant to the content of the manuscript.

(II) Highlight any revisions to the manuscript, so editors and reviewers can
see any changes made.

(III) Provide a cover letter to respond to the reviewers’ comments and explain, point by point, the details of the manuscript revisions.

(IV) If the reviewer(s) recommended references, critically analyze them to ensure that their inclusion would enhance your manuscript. If you believe these references are unnecessary, you should not include them.

(V) If you found it impossible to address certain comments in the review reports, include an explanation in your appeal.

Reviewer 2 Report

Comments and Suggestions for Authors

The manuscript is very interested for the effect of injection molding design with polymer viscosity. I would like to suggest author to revise manuscript as following below;
1. Abstract should be revise by adding the effect of polymer type (PP and PP composited) and their viscosity to complete the sentence and more attractive. 
2. Introduction should be adding more information of polymer processing modelling such as POLYFLOW for heat transfer and defect of injected product which were studied in many previous works. 
3. Experimental and Results :
- What is the tensile strength of injected plastic by microcellular injected? Please compare with conventional injected.  
- The results in Fig. 7-14 should be revised with comparative graph.

Comments on the Quality of English Language

The manuscript must be approved by english native speaker. 

Author Response

(The authors gave the same response as above.)

Round 2

Reviewer 2 Report

Comments and Suggestions for Authors

I would like to thank you very much. I agree with your revised manuscript.

Comments on the Quality of English Language

The manuscript can be improved by native speaker.

Author Response

Comments and Suggestions for Authors

Summary: As the title suggest the authors study Impact of Runner Size, Gate Size, Polymer Viscosity, and 2 Molding Process on Filling Imbalance in Geometrically balanced Multi-cavity Injection molding. The authors use PP and PC and their mixtures with glass fiber to study the differences in flow length and flow length deviation. The authors have designed a custom-made mold to produce ASTM D638 samples for analysis. Later on in this study they also use simulations and modelling to explain their results.

Overall Thoughts:  The authors have presented the results and discussion in a very clear manner. Most conclusions are supported by data very over-speculation. However, I have following suggestions for the authors to improve the current manuscript.

Dear Reviewer: Thank you for your insight comments and suggestions. My answer is as follows;

Suggestions/Correction:

Abstract. Remove line 16 and replaced imbalanced in line 18 with balanced as the context in this part of the abstract is extremely confusing.

Ans: 1. Line 16 was deleted and imbalanced was replaced with balanced. Thank you for suggestions.

Introduction. The introduction provides enough background to the current work done; however, it fails to identify the current knowledge gap and hence the need for the current study. I recommend separating the knowledge gap out in a separate paragraph and then have a paragraph outlining the work in this study.

Ans:  A new paragraph was added after line 111 and new sentenses were added “To date, few studies have examined the effects of polymer types, runner and gate sizes, and molding methods (such as CIM and MIM) on the filling imbalance problem in injection molding. This study employed both CIM and MIM processes”

Methods. Please add info on how the flow length were measure and how the Flow length deviations were calculated.

Ans: The flow length measurement was done by measuring from end the gate to the longest length (although melt front shape was not regulat). Flow length deviation was caluate by flow length minus average value by excel software.

Results and Discussion.

-        Line 227, I think the authors are referring to figure 4.a here. Several other instances of incorrect mentions of the figure number were observed. Please rectify this.

Ans:  Yes, Thank you great insight. Fig. 3a is replaced with Fig. 4a.

-       

Figure 7- 14. Please also label with the type of injection molding was used, below the runner length and gate size. 

-        I would suggest pairing the figures with similar conditions but different gate size. For example, figure 7 as 7.a and Figure 8 as 7.b. This will allow better comparison between the gate size parameter and its effects. Please ensure the Y-axis of all the figures are same to allow better visual comparison.

-Ans: Agreed with your comment of putting small and big gate results together. Scale of mininum and maximum are all the same.   Please see text for detail.

  Line 256 and 257, the PP needs to be edited properly.

 Ans.: Thank you. PP format was revised.

-        Line 310, Add explanation on why the trend doesn’t hold true for MIM.

Ans: Typically, the weight of parts produced by the CIM can be maintained at a certain value (e.g., 1 wt%), while the weight of parts by the MIM process—which is more com-plex—varies significantly, sometimes reaching as much as 5 wt%, based on our observa-tions. Several parameters can influence cell growth (e.g., injection speed and packing pressure). Although these parameters were kept constant, weight variations remain sub-stantial in the MIM process, resulting in an irregular relationship between small and big gates.

-        Line 331, 338 à it needs to be 16 a, b

Ans: Yes, thank you for your corrections.

-        Figure 17, add a gradient scale for what each color mean    

Ans: It was revised.

Conclusions. The authors need to re-write conclusions.  The current set of conclusions are very generalized and certainly do not point out the various non-linear and novel observations from this study. Additionally, the conclusions do not indicate what the future studies should explore to further the current work.

Ans:  was revised as follows

  1. The Big gate exhibited a longer flow length than the small gate for both the CIM and MIM processes. For the flow length deviation, neat PP demonstrated a slight flow length deviation compared to the glass fiber-filled PP for both CIM and MIM processes. However, this pattern does not hold for PC and glass fiber-filled PCs in either the CIM or MIï¼­
  2. Glass fiber-filled polymers exhibited higher viscosity than neat polymers for both PP and PC
  3. Cavities experiencing over-packing situations demonstrated better elongation for both neat PP and glass fiber-filled PP in the CIM process, while the tensile strengths remained the same for types of PP.
  4. The results of moldflow analysis agree with the experimental results, indicating that moldflow analysis can reduce costs and mold manufacturing time, thereby reducing the preparation time for the injection molding process.
  5. Polymer foaming can disturb the temperature distribution of the polymer melt in the runner for the multi-cavity geometrically balanced mold.

References.  Consistent formatting is recommended. Some references are missing page numbers and in some cases DOI is missing. It is up to the authors, what needs to be included however, the format should be consistent across all the references.

Ans: DOI information of proceedings and journals from India are not avaliable. Page numbers are provided.

Comments and Suggestions for Authors

The manuscript is very interested for the effect of injection molding design with polymer viscosity. I would like to suggest author to revise manuscript as following below;

Thank you for your good insight suggestions. My answers are as follow,

  1. Abstract should be revise by adding the effect of polymer type (PP and PP composited) and their viscosity to complete the sentence and more attractive. 

Ans: These sentenses of “ Polypropylene (PP) is a widely used semi-crystalline polymer due to its favorable flow characteristics, including a high melt flow index and the absence of a need for a mold temperature controller. Additionally, PP exhibits good elongation and toughness, making it suitable for applications such as box hinges. However, its tensile strength is a limitation; thus, glass fiber is added to enhance this property. It is important to note that the incorporation of glass fiber increases the viscosity of PP” are added after Three critical factors in this process are the polymer, injection molding machines, and the pro-cessing molds.

  1. Introduction should be adding more information of polymer processing modelling such as POLYFLOW for heat transfer and defect of injected product which were studied in many previous works. 

Ans: Through the advance of computer technology, computer simulation of injection molding is widely used. Researcher or manufacturer can save time in molding making or defect predictation by computer simulation. García-Granada et al. [] used software to solve nanopattern replication of complex structures by injection molding is added into Introduction.

  1. Experimental and Results :
    - What is the tensile strength of injected plastic by microcellular injected? Please compare with conventional injected.  

Ans: Foamed injection molding was performed after had received reviewer comments. Tensile test was done one week after foamed injection molding experiment (let the sample be stable) . So Figures 13 and 14 are added after Figure 12 of CIM molding. Descriptions and comparison are in my manuscript.

- The results in Fig. 7-14 should be revised with comparative graph.

Ans: Figs. 7-14 was replot per reviewer #1 and your suggestions  and 8 Figures was reduced to 4 Figures by comparing small and big gates at the same Figure.
